# Differential Hatching, Development, Oviposition, and Longevity Patterns among Colombian *Aedes aegypti* Populations

**DOI:** 10.3390/insects13060536

**Published:** 2022-06-10

**Authors:** Andrea Arévalo-Cortés, Yurany Granada, David Torres, Omar Triana-Chavez

**Affiliations:** Group Biología y Control de Enfermedades Infecciosas, Universidad de Antioquia—UdeA, Calle 70 No. 52-21, Medellín 050010, Colombia; eresbey2@gmail.com (Y.G.); david.torresh@udea.edu.co (D.T.); omar.triana@udea.edu.co (O.T.-C.)

**Keywords:** *Aedes aegypti*, hatching, development time, longevity, survivorship curve, fecundity, mortality

## Abstract

**Simple Summary:**

*Aedes aegypti* is a mosquito that transmits viruses responsible for several diseases in humans, such as dengue, Zika, and chikungunya. It is crucial to study mosquito populations from different countries and regions because control of disease transmission with insecticides can be more effective if adjusted to each population’s characteristics. For this reason, we determined several features of mosquitoes captured in different cities of Colombia: Neiva, Bello, Itagüí, and Riohacha. These included the length of their lifespan, the number of eggs they lay, and the stages in which they die. We found specific patterns for each population. This knowledge will help control programs determine the optimal times to apply insecticides and make surveillance, as well as the type of insecticide used.

**Abstract:**

Dengue, Zika, and chikungunya are arboviral diseases for which there are no effective therapies or vaccines. The only way to avoid their transmission is by controlling the vector *Aedes aegypti*, but insecticide resistance limits this strategy. To generate relevant information for surveillance and control mechanisms, we determined life cycle parameters, including longevity, fecundity, and mortality, of Colombian *Ae. aegypti* populations from four different geographical regions: Neiva, Bello, Itagüí, and Riohacha. When reared at 28 °C, Bello had the shortest development time, and Riohacha had the longest. Each mosquito population had its own characteristic fecundity pattern during four gonotrophic cycles. The survival curves of each population were significantly different, with Riohacha having the longest survival in both males and females and Bello the shortest. High mortality was observed in mosquitoes from Neiva in the egg stage and for Bello in the pupae stage. Finally, when mosquitoes from Neiva and Bello were reared at 35 °C, development times and mortality were severely affected. In conclusion, each population has a unique development pattern with an innate trace in their biological characteristics that confers vulnerability in specific stages of development.

## 1. Introduction

*Aedes aegypti* is the principal vector of several arboviruses such as dengue [1,2], Zika [3,4], and chikungunya [5,6]. Dengue affects 390 million people worldwide, with an estimated 96 million infections per year associated with any level of disease severity [7]. In 2019, the Americas reported more than 3 million dengue cases, the most significant number recorded in the region [8,9]. For Zika, over 2 billion people live in areas conducive to its transmission, and in 2015 the infection spread rapidly in the American continent, with 26 countries reporting active transmission and significant outbreaks observed in Brazil and Colombia [3,10,11,12]. In the latter, the epidemic peaked in early 2016, with an incidence of 377.7 cases per 100,000 inhabitants [11,12]. Chikungunya originated in Africa and has caused epidemics in Asia, the Indian subcontinent, Europe, the Americas, and the Pacific Islands [5]. With no vaccine or treatment available, the only accessible strategy is vector control. Insecticides are the primary tool for this purpose, but resistance to this chemical control has been observed in mosquito populations [13,14,15,16,17].

The biological characteristics of vector insects, such as the life cycle, longevity, oviposition, and stages more susceptible to mortality, are fundamental knowledge for their surveillance and control. They can help determine the most vulnerable stage in which the vector population is less able to face environmental stress and control measures [18,19,20,21]. The study of these characteristics has been relevant not only in mosquitoes [19,20,22,23,24,25,26] but also in other insects of medical importance, such as blood-feeding bugs [27,28], tsetse flies [29,30], and sandflies [31,32]. These data have also been used in predictive models for insect population density, contributing to operational strategies to control these species effectively [30,33,34,35]. Enrichment of data concerning disease vectors’ biological characteristics can ensure reliable model predictions of insect population dynamics [35].

*Aedes aegypti* is a holometabolous insect with immature stages (egg, four larval stages, and pupa) that develop in water and adults that live in terrestrial habitats [36]. The ability of *Ae. aegypti* to vary its biology and adapt to different environmental conditions can influence population densities, favoring its permanence in areas with virus transmission [37,38]. Thus, *Ae. aegypti* populations from different regions may present diverse growth and reproduction patterns [38]. In pathogen vectors, there are biological patterns inherent to each population, such as development, survival, mortality, longevity, and fertility, influenced by intrinsic and extrinsic factors specific to the region of origin [38,39,40,41,42]. Examples of intrinsic population factors are genetics [43,44,45], larval density [46,47] and microbiota composition [48,49,50,51], while extrinsic factors include temperature [47,52,53,54], water pH [55,56], presence of insecticides [57,58,59,60], and availability of food and biological resources [46,47,61,62].

This study aimed to compare the hatching, development time, eggs per female, longevity, survivorship curve, and mortality patterns of *Ae. aegypti* from different regions of Colombia to try to understand the population dynamics of each area. Moreover, we determined the effects of temperature on specific developmental stages in two mosquito populations, which showed different mortality and life expectancy patterns. The results will contribute to assessing and reducing the risk of *Ae. aegypti* infestation by local control strategies.

## 2. Materials and Methods

### 2.1. Study Site and Mosquito Collections

Immature stages and *Ae. aegypti* adults were collected between 2016–2018 in four Colombian cities (Figure 1) with assistance from staff involved in vector-borne disease programs from each municipality. Mosquitoes were sampled in 10–20 randomized houses from four neighborhoods in each city.

These cities were selected because they have had dengue epidemics within the last ten years. Riohacha is the capital of the La Guajira department and lies in the northern Caribbean coast, in the natural Guajira Peninsula subregion, within Ranchería river’s delta. It has an altitude of 2 m above sea level (m.a.s.l.) with an average temperature of 28 °C (minimum of 21 °C and maximum of 35 °C), considered a hot arid climate with average annual rainfall between 500–1000 mm [63]. From 1999 to 2010, La Guajira has presented seasonal dengue outbreaks every four years. Riohacha has concentrated 34.7% of these cases [64], with an incidence between 45.3 to 206.8 cases per 100,000 inhabitants [44]. Bello has an altitude of 1250 m.a.s.l. with an average temperature of 23 °C (minimum of 16 °C and maximum of 33 °C), classified as a warm semi-humid climate with an annual rainfall of 1500 to 2000 mm [63,65]. In this locality, between 2013 and 2019, there was an incidence of dengue cases per 100,000 inhabitants between 16.3 and 381.4, the highest being registered in 2016 [66]. Itagüí has an elevation of 1550 m.a.s.l. with a warm wet climate. It has an average temperature of 19.6 °C (minimum of 16 °C and maximum of 22 °C) and an annual rainfall of 2000 to 2500 mm [63,65]. In this city, the dengue cases per 100,000 inhabitants between 2013 and 2019 were 19.9 to 1063.5, with 2016 being the year with the highest incidence [66]. Neiva lies between the Central and Eastern Andes Cordilleras, in the natural Alto Magdalena valley subregion. It has an altitude of 442 m.a.s.l., an average temperature of 28 °C (minimum of 22 °C and maximum of 33 °C), and an annual rainfall of 1000 to 1500 mm, which is classified as a warm semi-arid climate [63]. The incidence of dengue between 2013 and 2019 was 164.3 to 1530.5 cases per 100,000 inhabitants; 2014 and 2019 were considered to be epidemic years with incidences of 1081 and 1530.5 per 100,000 inhabitants, respectively [67].

The Rockefeller strain was used as a control because it is considered a strain of vigor control for *Aedes aegypti*. It has been used as a reference strain in studies of insecticide resistance, microbiota and fitness cost under a genetic background with *kdr* mutations [68,69,70].

### 2.2. Rearing and Maintenance of Ae. aegypti Populations

Collections were reared under controlled conditions: temperature (28 ± 1 °C), relative humidity (80 ± 5%), and photoperiod (12 h light: 12 h dark). Mosquitoes were identified by the patterns of typical white lyre-shaped markings scales on the thorax [71]. For the identification of the larval stages of *Ae. aegypti* we followed the recognition method of Christophers [72], which is based in distinguishing changes in chaetotaxy and other external characters, such as greater branching of the hairs, greater number and branching of the spines and greater complexity of the respiratory siphon, the anal segment and the mouth parts. This was complemented with specific characteristics for each stage of larval development, such as the presence of the egg-breaker for L1, a transverse diameter of the head of approximately 0.3, 0.45, 0.65 and 0.95 mm for L1, L2, L3 and L4, respectively, a dark head for L3 (the head should never go completely dark for other instars), and the rudiments of the pupal respiratory trumpets for L4 [72]. F2 eggs were used after a storage time between 2 and 3 weeks. One hundred eggs were submerged in 50 mL of distilled and sterile water with 0.03 mg per larva of Tetramin^®^ fish food (Tetra, Melle, Germany), in a cup of 5 cm in diameter and 3 cm in height, until hatching. Larvae were subsequently reared in dechlorinated tap water in a plastic pan (8.5 cm in diameter and 11 cm in height) covered with muslin cloth at a density of 1 larva/mL, within the range used to optimize rearing of *Ae. aegypti* (0.6–2.8 larvae/mL) [73,74,75]. The pans were checked daily at the same hour of the day. Debris was removed using plastic pipettes, and water lost through evaporation was replaced to maintain constant levels. Four to six replicates were performed. All stages of larvae were subsequently fed daily with finely ground Tetramin^®^ fish food until pupation, with an increasing feeding regime for each stage per day consisting of 0.06 to 0.12 mg per larva in L1-L2, 0.24 mg in L3, and 0.48 mg in L4, following the strategy described by Araújo et al. [76]. All cohort individuals were sexed after emergence (passing through pupae to adult, male or female) and introduced into rearing cages (20 cm × 20 cm × 20 cm). Adults were provided with a 10% sterile sugar solution dispensed through a sterile cotton ball in a petri dish. Starting seven days after emergence, every week females were fed after 18 h of starvation with human O-positive blood through a piece of desalted porcine intestine used as a membrane for 60 min using water-jacketed glass feeders with a circulating system that maintained the blood at 37 °C. After each feeding, the engorged females were counted. Seventy-two hours after blood feeding, females were provided with an oviposition container (7.5 cm in diameter and 9 cm in height) with filter paper as substrate and 100 mL of water. They were allowed to oviposit for 72 h. The number of eggs laid per female during four gonotrophic cycles (GCs) was counted under a dissecting microscope. Gonotrophic cycle (GC) duration was determined as the time elapsed from blood feeding to oviposition [77]. The mortality of males and females was recorded daily until the death of all mosquitoes.

### 2.3. Effects of Temperature on Immature Stage Mortality

Because we observed changes in mortality and life expectancy associated with the pupal and egg stages in the Bello and Neiva populations, respectively, we wanted to know if this observation could be explained by temperature, which is one of the most critical variables that can modify these biological features in insect populations [47,78]. Thus, we submerged a cohort of 100 eggs at 21 ± 1 °C, 28 ± 1 °C and 35 ± 1 °C in controlled rearing conditions, as described above. The mortality of eggs, larvae (L1-L2, L3, L4), and pupae were registered daily. Four to six replicates were performed.

### 2.4. Biological Features and Statistical Analysis

Development periods, survivorship and mortality for eggs, larvae (L1-L2, L3, L4), and pupae were reported daily. The egg hatch rate was calculated by dividing the number of L1-L2 by the number of eggs at the beginning (cohort of 100 eggs per replicate) [56,79]. The development time for immature stages is shown as eclosion (egg), L1-L2 to L3, L3 to L4, and L4 to pupae. The time to pupation is the mean development time from L4 to pupae. The pupation rate was calculated by dividing the number of pupae by the number of eggs [80,81].

For adults, time of emergence (period during which the pupa matures into an adult), emergence rate (calculated by dividing the number of emerging adults by the initial number of eggs) [80], longevity, survivorship curves, and fecundity were also registered.

Both longevity and survival are related to mortality but reflect two completely different aspects. The mean longevity of a population indicates the average age that an individual can reach at death [82], while survival evaluates the average longevity of a population based on the number of individuals that have survived at different ages. Age-specific survivorship (lx) represents the proportion of individuals alive at a certain age from the starting number; see Equation (1). The survival curve graphs the values of lx as a function of age [83].

Fecundity was estimated based on the average number of eggs laid per female [21,40]. The sex ratio was determined by dividing the number of adult females by the number of adult males [42].

Various attributes of a horizontal life table were calculated applying the following equations [22,42,77,83,84,85,86]:(1)$$lx=Nx-N_{0}\;$$
where Nx is the number of survivors registered at the beginning of the age interval (x), and N_0_ is the starting number of mosquitoes in the population.

The proportion of individuals who die during the age interval (x) to (x+1), defined as (dx) is calculated as:(2)dx=lx−lx+1
where lx+1 is the proportion of surviving organisms registered at the beginning of the following age interval.

Age-specific mortality rate (qx), calculated as follows:(3)qx=dx / lx

Killing power (kx), is calculated as:(4)kx=Log Nx−Log Nx+1
where Nx+1 is defined as the number of survivors registered at the beginning of the following age interval.

The average of the probability of survival between two successive ages (Lx) is recorded as:(5)Lx=(lx−lx+1) / 2

The total number of days remaining to live for survivors of the cohort (Tx), when it reaches age x until the last member dies at age m, is estimated as:(6)$$Tx=\sum_{m}^{x}Lx$$
where “m” represents the maximum age reached

Life expectancy (ex) is defined as the mean number of days remaining to the survivors at age x:(7)ex=Tx /lx

Descriptive statistics were performed for all variables. To obtain multiple comparisons of the groups, an analysis of variance (One-way ANOVA) was conducted, and a Bonferroni post-hoc test (*p* < 0.05) was used to determine whether groups were significantly different. Additionally, a Student’s *t*-test was used (*p* < 0.05). These analyses were carried out using IBM SPSS Statistics Software (version 25.0 for Windows, SPSS Statistics Software, Armonk, NY, USA, IBM Corporation).

Adult survivorship was analyzed using the Kaplan–Meier survival analysis, and survivorship curves were generated. The Log-rank (Mantel–Cox) test was performed to compare the survival curves and establish statistically significant differences between mosquito populations. There were different levels of comparison between the populations and the Rockefeller control strain: overall pooled, overall strata (by sex: males and females) and pairwise strata were analyzed. All these analyses were performed using GraphPad Prism Software (version 5.1 for Windows, GraphPad Software, La Jolla, CA, USA, www.graphpad.com) and IBM SPSS Statistics Software (version 25.0 for Windows, SPSS Statistics Software, Armonk, NY, USA, IBM Corporation).

## 3. Results

### 3.1. Differences in Immature Stage Development Times, Egg Hatching, Pupation, and Emergence Rates in Ae. aegypti Populations

The mean hatching period was 1.22 ± 0.61, 1.39 ± 1.49, 1.93 ± 1.81, 2.49 ± 3.3, and 2.19 ± 2.33 days for Bello, Neiva, Itagüí, Riohacha, and Rockefeller, respectively (Figure 2A and Appendix A). The eclosion times between *Ae. aegypti* populations from Bello and Neiva were not statistically different, but both were significantly shorter than those from Itagüí and Riohacha (*p* < 0.05) (Figure 2A and Appendix A). In addition, these last two mosquito populations were different (*p* < 0.05) (Figure 2A and Appendix A).

The egg hatching rate for Neiva (89.75 ± 2.22) was significantly lower than for the other *Ae. aegypti* populations (Figure 2B).

The development time to reach L3 from L1 + L2 was different for each population (*p* < 0.05) (Figure 2A and Appendix A). The populations with the lowest and highest development times to reach this immature instar were Bello and Riohacha, respectively (Figure 2A and Appendix A).

There were statistically significant differences between all populations in the days necessary to pass from L3 to L4 (*p* < 0.05) (Figure 2A and Appendix A). The shortest time was recorded for Bello and the highest for Riohacha (Figure 2A and Appendix A).

Overall, the total larval development time was significantly different among all studied populations (*p* < 0.05), with the lowest for Bello (3.17 ± 1.03 days) and the longest for Riohacha (10.40 ± 6.03 days) (Appendix A).

The time to pupation was different between all mosquito populations, including the control Rockefeller (*p* < 0.05) (Figure 2A and Appendix A). In this stage, Neiva took significantly longer than Bello, Riohacha, and Itagüí (Figure 2A and Appendix A). The pupation rate from the initial number of eggs was considerably lower in mosquitoes from Neiva (85.50 ± 1.73) compared to those from Riohacha and Rockefeller (*p* < 0.05) (Figure 2C).

The male and female emergence times for mosquitos from Bello were significantly shorter than those from other populations (*p* < 0.05) (Figure 3A and Appendix A). Furthermore, Neiva males took a substantially higher time to reach adulthood than Bello and Itagüí, but not Riohacha (*p* < 0.05) (Figure 3A and Appendix A). This same population had significantly higher times for emergence of female adults than all others (*p* < 0.05) (Figure 3A and Appendix A). Additionally, for Bello, Neiva, Itagüí, and Rockefeller, within each population, males took significantly fewer days to emerge than females (these differences are not marked in Figure 3A). In contrast, for Riohacha, males and females needed similar times (Appendix A). The latter population had a significantly higher emergence rate than the others (Figure 3B).

The average period in days from egg to adult of the studied mosquito populations is shown in Appendix A, where Bello had the lowest life cycle duration for males and females. In contrast, Riohacha had the highest for both (Appendix A).

### 3.2. Life Table Attributes of Immature Stages in Ae. aegypti Populations

The horizontal life table of immature stages is shown in Appendix A. The highest mortality, represented by *qx* and *kx*, occurred in eggs, in which Neiva mosquitoes had a significant difference with the populations from Bello, Itagüí, Riohacha, and Rockefeller (Figure 4A,B and Appendix A). The other significant difference in *qx* occurred in pupae between Bello and the control (Figure 4A and Appendix A). The mortality found in these stages is consistent with the low life expectancy (*ex*) observed in eggs and pupae in the same two populations (Figure 4C and Appendix A). Life expectancy gradually decreased throughout development for all populations of *Ae. aegypti* (Figure 4C and Appendix A). Although Bello had a significantly lower life expectancy than other populations in the larval stages, this was not mainly reflected in the corresponding mortality (Figure 4A–C and Appendix A).

### 3.3. Sex Ratio, Longevity, and Survivorship of Adult Ae. aegypti Populations

The life table attributes of adult populations obtained in this study are shown in Appendix A. There were no statistically significant differences between the sex ratios of any of the populations (Appendix A) but significant differences in longevity of females and males were observed in all of them (*p* < 0.05) (Appendix A). Riohacha had the most extended longevity in both genders, and Bello had the shortest (Appendix A). Likewise, the female adult longevity of all populations was significantly more extended than that of males (* *p* < 0.05) (Appendix A).

The survival for males was significantly shorter than for females when analyzing overall pooled populations Chi-square (*Χ*^2^): 1073.63 df: 4 (*p* < 0.05). In males, Riohacha had the most prolonged survival and Bello the shortest. Bello’s male survival curve showed a rapid decrease on day 11, while Riohacha’s showed a gradual decline until day 58, and the curve dropped on day 73 (Figure 5A). Bello had the shortest survivorship curve in females, which decreased rapidly on day 15. In contrast, Riohacha had a more stable survival curve, which descended gradually on day 71 and then fell on day 85 (Figure 5B). Appendix A shows the significantly different values of the survival curves when pairwise strata (sex) comparison is made between the populations using a Log-rank (Mantel–Cox) test.

The life expectancy of adult *Ae. aegypti* females was higher than males and gradually decreased for both sexes with increasing age (Appendix A).

### 3.4. Fecundity Profile of Ae. aegypti Populations

The fecundity of the first four female gonotrophic cycles (GCs) displayed significant differences among the populations studied, showing a specific profile for each one (Figure 6). This variability in fecundity between populations occurred during the GCs, even though females had the same age. The number of eggs laid by mosquitoes from Neiva was low and remained constant throughout the four GCs, with 20.05 to 23.30 eggs per female. This resulted in significant differences to at least one population in each GC (Figure 6). Bello had a range of eggs per female between 30.57 and 51.73. Its fecundity was significantly higher than the other populations in the first GC; the number of eggs per female remained high in the second and third GCs with no significant differences to Itagüí and Riohacha and dropped in the fourth (Figure 6). Itagüí had GCs alternating between low and high fecundity, with a range of 17.10 to 44.84 eggs per female (Figure 6). For Riohacha, the first GC was low, like Neiva and Itagüí, but from the second to fourth GCs, the fecundity was stable, and the mean eggs laid per female were ≥42.21 (Figure 6).

### 3.5. Effect of Temperature on Immature Stage Development Times, Egg Hatching, Pupation, and Emergence Rates for Bello and Neiva

To assess the effect of temperature on immature stages of mosquito development, we reared *Ae. aegypti* from Bello, Neiva, and the control Rockefeller at 21, 28, and 35 °C. For Bello, we found that the times for eclosion and development from L1 + L2 to L3 and L3 to L4 were significantly shorter at 28 °C than at 35 °C and 21 °C and that these times were significantly longer at 35 °C than at 21 °C (Figure 7A–C and Appendix A). For Neiva and Rockefeller, we also found these times considerably shorter at 28 °C. At 35 °C, mosquitoes from Neiva had longer times substantially only in eclosion and L1 + L2 to L3, while no significant differences were observed in L3 to L4 compared with 21 °C (Figure 7A–C and Appendix A).

As for pupation time (L4 to pupae), we found significant differences for Bello and Rockefeller in the three temperatures evaluated, with 28 °C showing the shortest pupation time, followed by 35 °C and 21 °C. Meanwhile, Neiva showed no differences between 28 °C and the other temperatures and only had a significantly longer time at 21 °C in relation to 35 °C (Figure 7D and Appendix A).

The hatching and pupation rates for Bello, Neiva, and Rockefeller showed similar patterns. Within the same population, both features were higher at 21 °C and 28 °C, without significant differences, while the lowest rates were observed at 35 °C. This effect was more significant in the Neiva population (Figure 8 and Appendix A).

Male and female emergence times for Bello and Rockefeller were significantly lower at 28 °C than at 21 °C and 35 °C. Meanwhile, for Neiva, there were no differences between the temperatures evaluated (Figure 9A,B and Appendix A). For all populations, the emergence rate was not different between 21 °C and 28 °C, but was significantly lower at 35 °C, with Neiva showing a pronounced drop (Figure 9C and Appendix A).

### 3.6. Effect of Temperature on Mortality in Immature Stages from Bello and Neiva

The horizontal life tables of the immature stages for Bello and Neiva reared at 21, 28, and 35 °C are shown in Appendix A. Temperature significantly altered the mosquito mortality rate (*qx*) of egg and L1 + L2 stages (Figure 10 and Appendix A). At 35 °C, these stages had a significantly higher mortality rate than at 28 °C and 21 °C for Bello, Neiva, and Rockefeller (Figure 10A–C and Appendix A). Previously, we observed that at 28 °C, mortality was associated with the egg stage in Neiva (Figure 4A). However, when we increased the temperature to 35 °C, we observed that the mortality rate at the egg stage for Neiva had a significant increase up to 0.83 on average, more than double that of the other populations. This indicates that the most vulnerable stage for Neiva is affected by temperature increases (Figure 10B and Appendix A). We also observed at 28 °C that mortality rate was associated with the pupae stage in Bello (Figure 4A). However, for this case, the increment in temperature to 35 °C did not affect the mortality significantly (Figure 10A and Appendix A). Additionally, at 35 °C, life expectancy (*ex*) was lower in the stages with higher mortality rates in all populations (Appendix A).

## 4. Discussion

Egg and early stages of insect development are the most vulnerable to biotic and abiotic factors in the field, and they show high mortality rates [18,83]. In this study, egg development times ranged from 1.22 to 2.49 days, with the populations from Bello and Neiva being the fastest to hatch (Table 1). In addition, high mortality was associated with the egg stage in Neiva, and as a result, its hatching rate was significantly lower (Table 1). A plethora of studies from different geographic regions carried out under conditions similar to ours reveals variability in egg parameters. In Brazil, *Ae. aegypti* populations from different bioclimatic regions differed significantly in the duration of embryonic development and egg viability, with means ranging between 3.8 and 4.4 days and 58% to 84%, respectively [38]. In an *Ae. aegypti* population from Peru, egg development of one day on average was reported with a *qx* range between zero and 0.08, while a population from Saudi Arabia had an egg development of 5.3 days and a hatching rate of 72% [22,77]. In other species such as *Culex quinquefasciatus,* populations from different climatic regions of India had significant differences in egg hatching period (range 1.7–2.4 days), hatching rate (range 80.5–95.6%), and mortality rate (range of 4.4–19.5%) [42]. These studies show that egg development and mortality can vary according to geographical origin.

Regarding fecundity, the population from Neiva had the least number of eggs oviposited per female throughout the four gonotrophic cycles (Table 1). Mosquitoes from Bello showed a fecundity curve, highest in the second and third gonotrophic cycles. Moreover, mosquitoes from Itagüí were characterized by the alternation of low and high fecundity cycles, while Riohacha′s fecundity was high and stable from the second gonotrophic cycle onwards. We used type O human blood as a food source, which has been shown to have an essential effect on fecundity [87,88]. The number of eggs per female is more significant when they are fed human blood [88], and type O blood has the best digestibility and is preferred by *Ae. aegypti* females [89,90]. Our work contributes for the first time that *Ae. aegypti* Colombian populations from different geographical origins fed with O-positive human blood present a characteristic pattern in fecundity. Another aspect to consider is the possibility that populations meet and cross in nature [91], which could impact their fecundity. Thus, future trials will be conducted by crossing the vigor control strain Rockefeller with Neiva to verify if this could have an impact on egg fecundity and mortality.

Larval and pupal development times were also different between the *Ae. aegypti* populations studied (Table 1). Larvae from Neiva, Itagüí, and Riohacha had prolonged development in relation to Bello, while time to pupation was longer for Riohacha, Itagüí, and Neiva, in ascending order. Mortality at the larval stage was low in all populations. Bello had significantly higher mortality than the control at the pupae stage (Table 1). Similar differences have been reported in *Ae. aegypti* populations from different geographic regions of Brazil, with a larval development time of 6.3–8.3 days with a viability of 92–100%, and a pupation time of 2–2.5 days with a viability of 63–94.5% [38]. Additionally, in *Cx. quinquefasciatus* there were differences in larval and pupal development and mortality among populations of different climatic sources [42]. Our finding of prolonged times in both larvae and pupae development in three populations is not rare since, for example, larval development times in populations of *Ae. aegypti* with ranges from 9.15 to 10.89 days have been reported, similar to what we found for Riohacha [37]. These long development times may represent a disadvantage since larvae and pupae are exposed longer to predators and other factors that can eliminate them and, therefore, may influence the behavior of these populations in the field [92,93,94].

The survival in mosquitoes from Bello was the shortest both in males and females, while Riohacha was the longest-lived population. Female survival was shortest in Bello, followed by Neiva, Itagüí, and Riohacha. The longevity times we observed were longer than those reported for *Ae. aegypti* strains from other geographical regions, such as those from Argentina (males: 7.3–8.8 days; females: 11.5–58 days), Saudi Arabia (males: 9.59 days; females 17.14 days), and Brazil (males: 14.22 to 32.06 days; females: 13.81–30.08 days) [41,52,77,95]. Our results indicate that there are factors that negatively affect the survival of mosquitoes from Bello, Neiva and Itagüí. The composition of the gut microbiota may be one of these factors, as we have previously described specific patterns for these populations [96]. For the Neiva population, insecticide resistance is another factor that may be involved in this phenomenon [96]. Conversely, the Riohacha population showed higher survival, especially in males, indicating adaptation to this region’s hot climate [97]. Further studies should analyze this population’s fitness in order to discern which factors are most important for mosquito survival.

Prolonged longevity in females may represent a risk factor for dengue virus spread [95] because transmission will only happen if the mosquito lives longer than the viral extrinsic incubation period [98,99,100]. Furthermore, longevity can influence fecundity, as it is expected that during a female’s lifespan they will be able to complete at least two gonotrophic cycles [95,101,102,103]. In our study, the population with the lowest longevity (Bello, with 39.9 days) had a fall in fecundity in the fourth gonotrophic cycle. Fecundity tends to decrease in females close to their decline in life [101,102]. During long lifespans, oviposition patterns may show several peaks as we saw in the populations from Riohacha and Itagui. Fecundity peaks during a female’s lifespan have been reported in *Aedes spp.* and other arthropods, indicating that both longevity and fecundity are important in mosquito population dynamics [101,104,105,106,107].

Several of our findings may be partly explained by differences in the microbiota [96], as bacteria associated with mosquitoes can influence development in several ways. For example, bacteria present in breeding sites and the metabolites they produce are essential for *Ae. aegypti* egg hatching [108,109] and eggs infected with *Wolbachia* show delayed embryo development, delayed eggshell formation, and a diminished egg hatch rate [110,111,112]. It has also been determined that gut-associated bacteria are necessary for egg production in *Ae. aegypti* and *Ae. atropalpus* [113] and that *Anopheles coluzzii* infected with *Chromobacterium violaceum* lays significantly less eggs [114]. Although *Ae. aegypti* larvae can develop in axenic (bacteria free) conditions, the time to pupation is longer and pupal development is stunted compared with populations reared in the presence of all bacterial communities or a single microorganism, suggesting that bacteria′s main role is nutritional [115]. Furthermore, *Ae. aegypti* has plasticity to adapt to the microorganisms present in the diet, which in turn affects mosquito development. For example, when *Asaia* and *Escherichia coli* were used as diet, time to pupation decreased and larvae had a high survival span [116]. Additionally, the data obtained in this study provides relevant information for the release of *Ae. aegypti* populations infected with *Wolbachia,* especially in Bello and Itagüí, where such a vector control program is now running [117].

Our previous works have established that the Neiva and Riohacha populations have a lambda-cyhalothrin resistant phenotype associated to *knock-down resistance* (*kdr*) mutations in the voltage-gated sodium channel gene [13,96,118]. Meanwhile, the Bello and Itagüí populations have a very low frequency for the mutant alleles and are susceptible to lambda-cyhalothrin [13,96,118]. The insecticide resistance status of each population is a factor that may influence the life cycle parameters analyzed in this study. For example, field and laboratory *Ae. aegypti* populations resistant to pyrethroids show a decreased egg hatching rate and resistance to desiccation under continuous insecticide exposure or even in the absence of insecticide pressure [59,119,120]. This effect is more evident when the time in contact with the insecticides is longer and increases through generations [59,120]. Furthermore, prolonged larval development times have been reported in *Ae. aegypti* populations that carry the V1016G/S989P genotype and show high resistance to lambda-cyhalothrin and low-level resistance to deltamethrin [121], as well as in organophosphate-resistant *Ae. aegypti* populations [122]. In agreement with these findings, we found that the Neiva population, which has the highest allelic frequency rate for *kdr* mutations, has high mortality at the egg stage, low hatching rate and prolonged larval development and pupation times. The phenotypic and genotypic variability associated with insecticide resistance in Colombian *Ae. aegypti* populations [118] may be generating this wide spectrum of biological characteristics; future studies will be required to support this.

Temperature has been described as one of the main extrinsic factors that influences *Ae. aegypti* life cycle parameters [47,78] because it plays a fundamental role not only in the biology of the vector, such as its development time and population density [47,52,123,124], but also in geographical dispersion [78,125] and arboviral transmission [97,126,127]. In *Aedes aegypti*, it has been described that the optimal development rates lie between 22 to 32 °C, with high mortality rates at the extremes (18 °C and 34 °C) [39,41]. The variability between mosquito populations concerning temperature is complex since the mosquito can have plasticity in the way it adapts to environmental conditions [128]. We found that 28 °C was the optimal temperature for all parameters evaluated (Table 2), although the egg hatching rate was significantly decreased for Neiva at this temperature (Table 1). Our results indicate that temperature affected our populations, raising mortality in specific developmental stages (Table 2). For the Neiva population, we identified a pattern associated with the egg stage as more vulnerable (Table 2). In that sense, mosquitoes from this locality may have an innate trace in their biological characteristics, making some stages more vulnerable to extrinsic factors such as temperature.

Several other studies have provided evidence that temperature can affect the life cycle of specific *Ae. aegypti* populations. For example, in Brazil, a significant interaction was observed between population origin and temperature effects on *Ae. aegypti* immature development times, as it was longer at 16 °C to 22 °C and decreased at higher temperatures (28 °C to 36 °C), in contrast to what was observed in our work at 35 °C [52]. Another study with an *Ae. aegypti* population from Paraná, Brazil, showed that the egg hatching rate can be close to 50% at temperatures of 5 °C and 25 °C, verifying that the mosquito can have adaptations at the local level, tolerating low temperatures [129]. Furthermore, in an *Ae. aegypti* population from Trinidad, West Indies, it was seen that the egg hatching rate was low (1.6%) at 35 °C, improved to 57% at 27 °C, and reached 98% at 25 °C [54]. In a similar fashion, Farnesi et al. studied the egg hatching rate between 16 and 36 °C and found the highest rates between 22 and 28 °C (range 93.3% to 96%), a lower rate at 35 °C (48.5%), and no hatching at 36 °C [123], comparable to this work. The above indicates that temperature and geographical origin interact to influence *Ae. aegypti* development time, mortality, and hatching rate.

## 5. Conclusions

Finally, another aspect to be considered is recent reporting of an *Aedes albopictus* increase in Colombia, since this species is an important competitor of *Ae. aegypti* [130]. Likewise, the release of mosquitoes infected with *Wolbachia* could be affecting the life tables of these mosquitoes. In conclusion, the data contained here is a contribution to fundamental knowledge that local vector control entities can consider for improving strategies by determining specific times for larvicide and adulticide application and *Ae. aegypti* monitoring.

## Figures and Tables

**Figure 1 insects-13-00536-f001:**
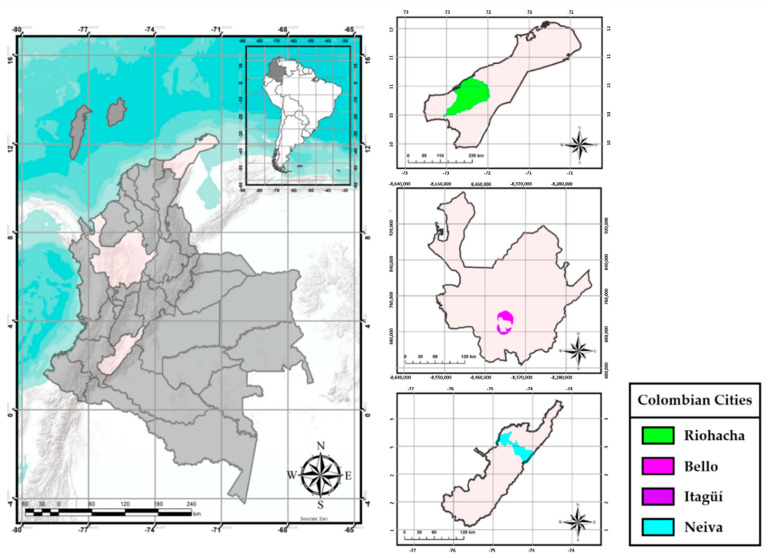
Map of Colombia showing cities where samples of *Aedes aegypti* mosquitoes were collected.

**Figure 2 insects-13-00536-f002:**
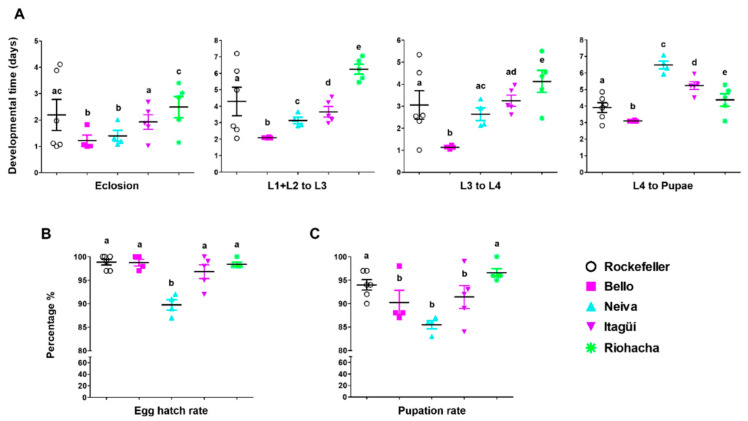
(**A**) Developmental time (days) of each immature instar, (**B**) egg hatch rate and (**C**) pupation rate of Colombian *Ae. aegypti* populations reared at 28 ± 1 °C temperature, 80 ± 5% relative humidity, and 12 h light: 12 h dark photoperiod. The mean and standard error of the mean were plotted. Each point represents a replicate from a cohort of 100 eggs. The total number of replicates were: Bello and Neiva (4); Itagüí and Riohacha (5), and control Rockefeller (6). L: larvae. The time to pupation is the mean development time from L4 to pupae. One-way analysis of variance (ANOVA) was conducted, followed by a Bonferroni post-hoc test to account for multiple comparisons between *Ae. aegypti* populations and the control (Rockefeller). Populations sharing the same letters are not significantly different (*p* > 0.05).

**Figure 3 insects-13-00536-f003:**
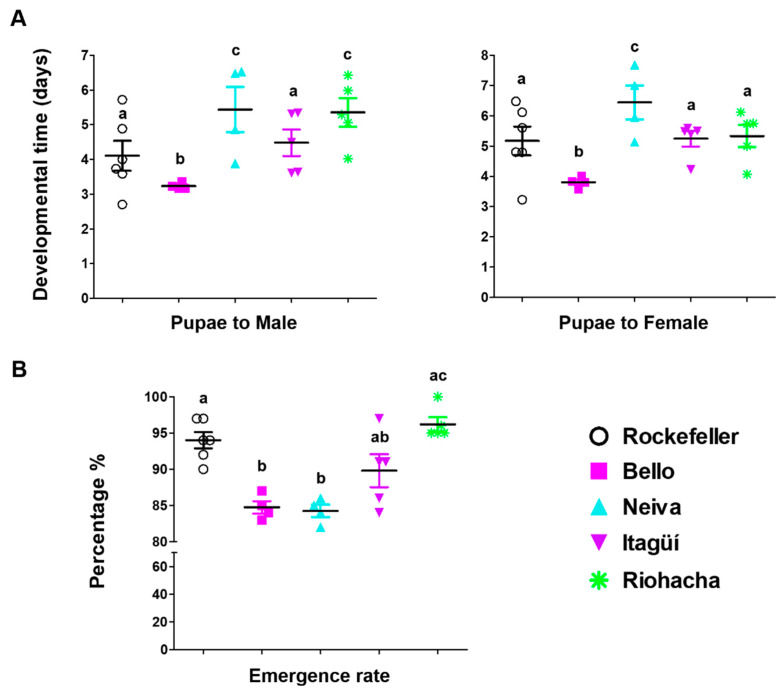
(**A**) Emergence time (days to develop from pupae to adult male or female) and (**B**) emergence rate of Colombian *Ae. aegypti* populations reared at 28 ± 1 °C temperature, 80 ± 5% relative humidity, and 12 h light: 12 h dark photoperiod. The mean and standard error of the mean were plotted. Each point represents a replicate from a cohort of 100 eggs. The total number of replicates were: Bello and Neiva (4); Itagüí and Riohacha (5), and control Rockefeller (6). A one-way analysis of variance (ANOVA) was conducted, followed by a Bonferroni post-hoc test to account for multiple comparisons between *Ae. aegypti* populations. Populations sharing the same letters are not significantly different (*p* > 0.05).

**Figure 4 insects-13-00536-f004:**
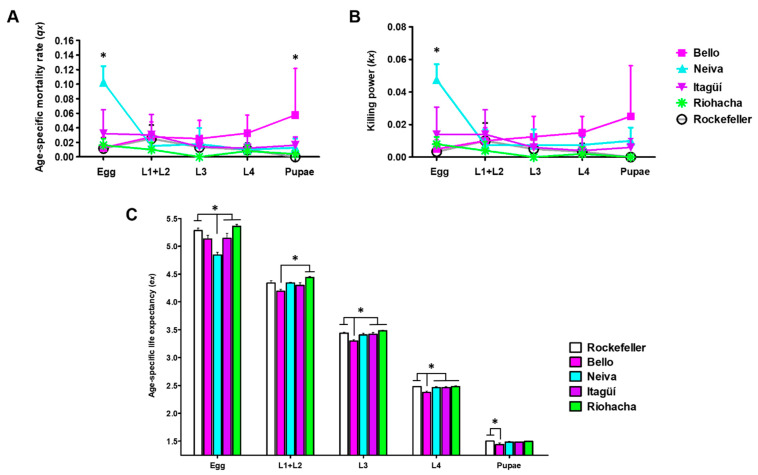
(**A**) Age-specific mortality rate (*qx*), (**B**) killing power (*kx*), and (**C**) age-specific life expectancy (*ex*) of Colombian *Ae. aegypti* populations reared at 28 ± 1 °C temperature, 80 ± 5% relative humidity, and 12 h light: 12 h dark photoperiod. Mean and standard deviation (SD) are shown. The total number of replicates were: Bello and Neiva (4); Itagüí and Riohacha (5), and Rockefeller control (6). Each replicate started from a cohort of 100 eggs. One-way analysis of variance (ANOVA) was conducted (* *p* < 0.05). A Bonferroni post-hoc test to account for multiple comparisons between populations of *Ae. aegypti* is presented in Appendix A.

**Figure 5 insects-13-00536-f005:**
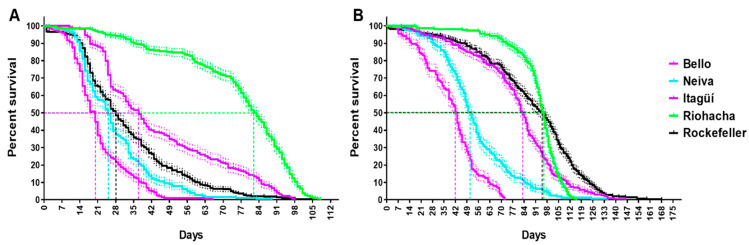
Percentage of survival for (**A**) males and (**B**) females of Colombian *Ae. aegypti* populations reared at 28 ± 1 °C temperature, 80 ± 5% relative humidity, and 12 h light: 12 h dark photoperiod. Kaplan–Meier plots were generated in GraphPad Prism. The mean and standard error (SE) are shown. The total number of replicates were: Bello and Neiva (4); Itagüí and Riohacha (5), and Rockefeller control (6). Each replicate started from a cohort of 100 eggs. Statistical differences were calculated by Log-rank (Mantel–Cox) test. Males: Chi square (*Χ*^2^): 499.9 df: 4 (*p* < 0.05). Females: *Χ*^2^: 620.0 df: 4 (*p* < 0.05). The dotted line represents the day at median survival for each population.

**Figure 6 insects-13-00536-f006:**
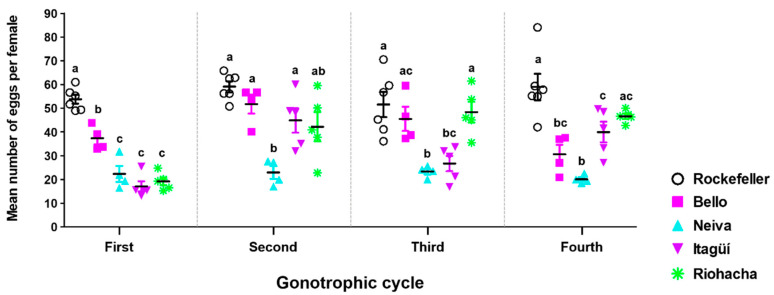
Fecundity per gonotrophic cycle of Colombian *Ae. aegypti* populations reared at 28 ± 1 °C temperature, 80 ± 5% relative humidity, and 12 h light: 12 h dark photoperiod. The mean and standard error of the mean were plotted. Each point represents a replicate that started from a cohort of 100 eggs. The total number of replicates were: Bello and Neiva (4); Itagüí and Riohacha (5), and control Rockefeller (6). One-way analysis of variance (ANOVA) was conducted, followed by a Bonferroni post-hoc test to account for multiple comparisons between *Ae. aegypti* populations. Populations sharing the same letters are not significantly different (*p* > 0.05).

**Figure 7 insects-13-00536-f007:**
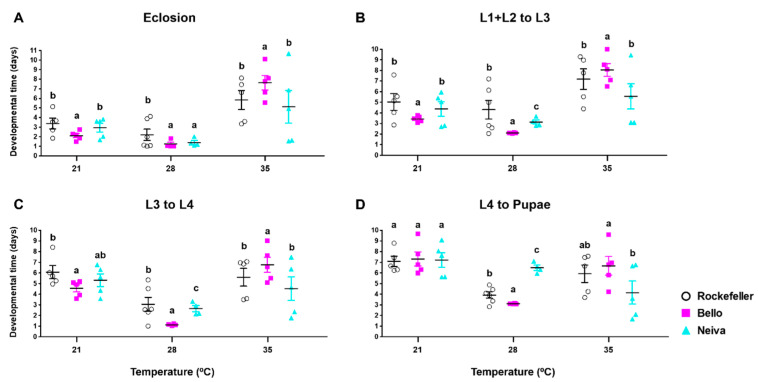
Developmental time (days) of each immature instar of two Colombian *Ae. aegypti* populations reared at 21 ± 1 °C, 28 ± 1 °C and 35 ± 1 °C temperatures, 80 ± 5% relative humidity, and 12 h light: 12 h dark photoperiod. (**A**) Eclosion, (**B**) L1 + L2 to L3, (**C**) L3 to L4, and (**D**) L4 to Pupae. The mean and standard error of the mean were plotted. Each point represents a replicate that started from a cohort of 100 eggs in each. The total number of replicates were: Bello and Neiva 28 ± 1 °C (4); control Rockefeller 28 ± 1 °C (6); Bello, Neiva and control Rockefeller 21 ± 1 °C and 35 ± 1 °C (5). L: larvae. The time to pupation is the mean development time from L4 to pupae. One-way analysis of variance (ANOVA) was conducted, followed by a Bonferroni post-hoc test to account for multiple comparisons between *Ae. aegypti* populations at the same temperature by age. Populations sharing the same letters are not significantly different (*p* > 0.05).

**Figure 8 insects-13-00536-f008:**
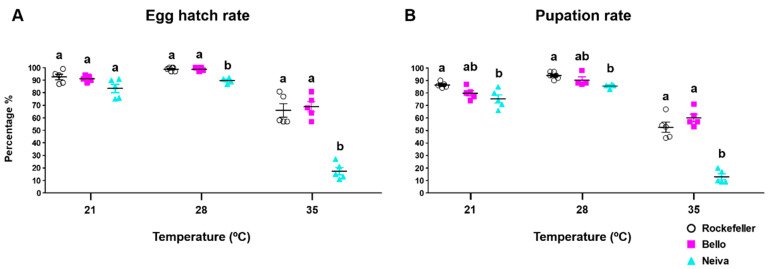
(**A**) Egg hatch rate and (**B**) pupation rate of two Colombian *Ae. aegypti* populations reared at 21 ± 1 °C, 28 ± 1 °C and 35 ± 1 °C temperatures, 80 ± 5% relative humidity, and 12 h light: 12 h dark photoperiod. The mean and standard error of the mean were plotted. Each point represents a replicate that started from a cohort of 100 eggs. The total number of replicates were: Bello and Neiva 28 ± 1 °C (4); control Rockefeller 28 ± 1 °C (6); Bello, Neiva and control Rockefeller 21 ± 1 °C and 35 ± 1 °C (5). One-way analysis of variance (ANOVA) was conducted, followed by a Bonferroni post-hoc test to account for multiple comparisons between *Ae. aegypti* populations at the same temperature. Populations sharing the same letters are not significantly different (*p* > 0.05).

**Figure 9 insects-13-00536-f009:**
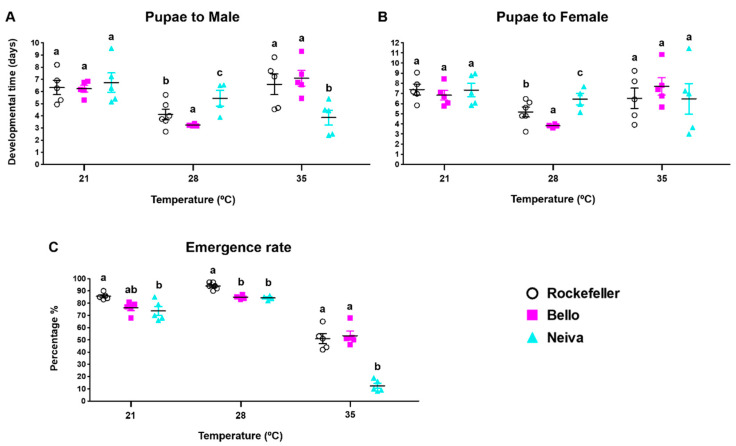
Emergence time, days to develop from pupae to male (**A**) or female (**B**) and emergence rate (**C**) of two Colombian *Ae. aegypti* populations reared at 21 ± 1 °C, 28 ± 1 °C and 35 ± 1 °C temperatures, 80 ± 5% relative humidity, and 12 h light: 12 h dark photoperiod. The mean and standard error of the mean were plotted. Each point represents a replicate that started from a cohort of 100 eggs. The total number of replicates were: Bello and Neiva 28 ± 1 °C (4); control Rockefeller 28 ± 1 °C (6); Bello, Neiva and control Rockefeller 21 ± 1 °C and 35 ± 1 °C (5). One-way analysis of variance (ANOVA) was conducted, followed by a Bonferroni post-hoc test to account for multiple comparisons between *Ae. aegypti* populations at the same temperature. Populations sharing the same letters are not significantly different (*p* > 0.05).

**Figure 10 insects-13-00536-f010:**
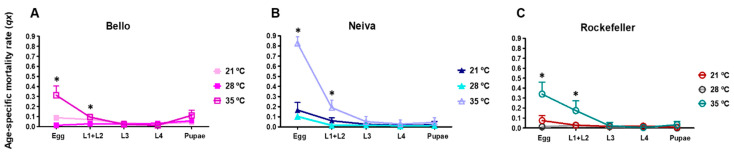
Age-specific mortality rate (*qx*) at 21, 28, and 35 °C, 80 ± 5% relative humidity, and 12 h light: 12 h dark photoperiod. (**A**) Bello, (**B**) Neiva and (**C**) Rockefeller. Mean and standard deviation (SD) are shown. One-way analysis of variance (ANOVA) was conducted (* *p* < 0.05). A Bonferroni post-hoc test to account for multiple comparisons between temperatures for *qx* within the same population and developmental stage is presented in Appendix A.

**Table 1 insects-13-00536-t001:** Summary of the results from four populations reared at 28 ± 1 °C temperature, 80 ± 5% relative humidity, and 12 h light: 12 h dark photoperiod. For each trait, only populations with extreme values that are significantly different are shown.

Trait	Lowest Value	Highest Value
Eclosion time	Bello and Neiva	Riohacha
Time from L1 + L2 to L4	Bello	Riohacha
Pupation time	Bello	Neiva
Female emergence time	Bello	Neiva
Male emergence time	Bello	Neiva and Riohacha
Egg hatching rate	Neiva	
Pupation rate	Neiva	
Emergence rate		Riohacha
Female longevity	Bello	Riohacha
Male longevity	Bello	Riohacha
Egg mortality		Neiva
Pupae mortality		Bello
Fecundity	Neiva	

**Table 2 insects-13-00536-t002:** Summary of the results from two populations reared at 21 ± 1 °C, 28 ± 1 °C and 35 ± 1 °C temperatures, 80 ± 5% relative humidity, and 12 h light: 12 h dark photoperiod. For each trait, only populations with extreme values that are significantly different are shown.

Trait	Lowest Value	Highest Value
Eclosion time	Bello and Neiva 28 °C	Bello and Neiva 35 °C
Time from L1 + L2 to L4	Bello and Neiva 28 °C	Bello and Neiva 35 °C Neiva (L3 to L4) 21 °C
Pupation time	Bello 28 °C Neiva 35 °C	Bello and Neiva 21 °C
Female emergence time	Bello 28 °C	
Male emergence time	Bello 28 °C	
Egg hatching rate	Bello and Neiva 35 °C	Bello and Neiva 21 and 28 °C
Pupation rate	Bello and Neiva 35 °C	Bello and Neiva 21 and 28 °C
Emergence rate	Bello and Neiva 35 °C	Bello and Neiva 21 and 28 °C
Egg mortality		Neiva 21, 28 and 35 °C Bello 35 °C
L1 + L2 mortality		Bello and Neiva 35 °C
Pupae mortality		Bello 28 °C

## Data Availability

The data presented in this study are available on request from the corresponding author. The data are not publicly available due to institutional privacy guidelines.

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
