# Peer review of "Differential Hatching, Development, Oviposition, and Longevity Patterns among Colombian Aedes aegypti Populations"

_insects, 2022, doi:10.3390/insects13060536_

Round 1

Reviewer 1 Report

I very much appreciate the opportunity to review this interesting manuscript on the "Differential hatching, development, oviposition, and longevity patterns among Colombian Aedes aegypti populations." I enjoyed reading the manuscript, and I would like to commend the authors for several key points including: 

1.         The use of appropriate sample sizes

2.         The detailed analysis of the mosquito developmental biology

3.         The use of appropriately sampling and data analysis approaches. 

These are all important strengths of the study. 

Considering these strengths, though, as I read the manuscript, I found some areas where I would have appreciated greater clarity. I believe the paper could be further strengthened by added information about: 

1.         Why did the author use Rockefeller strain as a control? 

2.         The methods used. After reading the methods section, I found myself wondering about some of the details of the methods used. By locating and reading the earlier publication on this work referenced in the manuscript, I answered some of my questions. However, I think it is unlikely that most readers would take the time to search out the companion publication. Without doing so, I feel that the reader may question the validity of the approach taken. I suggest expanding the description of this paper's methods (e.g., how or what method the author used to characterize each larval stage?). 

3.         Figures. I recognize that the figures present in this study are somewhat challenging to interpret. For example, in Fig 2A. for L4 to pupae bar category, why only compare Riohacha to Control but not the other. I highly encourage the author to put more emphasis on his/her figures. Keep all the figures consistent; for example, "ns" was used in Fig 3A but not in other Figures. 

4.         Results: I recommend that the author address each figure more carefully. For example, some of the bar graphs, such as Fig 2A and Fig 3A and B, did not clearly describe in the result section. 

5.         Discussion: There were many interesting results in this study. One in particular that caught my attention was the survival analysis. There is a factor or factors that negatively influence the survival of mosquitoes collected from Bello, Itaqui, and Neiva; It would be great if the author could discuss it more. Conversely, Riohacha seems to be very different from the other sites and the control (especially for the male). 

Reviewer 2 Report

The manuscript by Arévalo-Cortés et al. has an excellent and well-written story. It adds new and exciting data on various life cycle parameters such as longevity, fecundity, and mortality of Ae. aegypti populations from different geographical locations in Colombia. The outcome has potential implication in developing better insect management programs. I have a minor comment need to be addressed before the article gets accepted for publication

I encourage authors to re plot the figures in a scatter-plot or scatter-bar-plot manner so the readers can appreciate the distribution of individual data points. Also, indicate the replicate size per group in figure legends.

Reviewer 3 Report

About 50% of the world’s population is currently susceptible to a mosquito borne disease. The vaccine Dengvaxia is only approved for a limited population of these. In the face of growing insecticidal challenges to mosquito control, it is imperative to have a better understanding of both genomic and environmental factors which may impact vector control. Authors Arevalo-Cortes et al present a thorough investigation and comparison of Aedes aegypti populations from four different sites in Colombia - Neiva, Bello, Itagüí, and Rioha. Interestingly, the results suggest evolutionary implications and ecological adaptations of Aedes from different regions suited to a particular region. While significant differences were found in between the strains for factors involving longevity, time spent as larval instars and pupation,  to me the most interesting results were the impact of rearing temperature on the  various strains. Temperature and humidity play a crucial role in establishment as well as control of Aedes species in these regions, and this warrants further investigation. Overall, the work performed is thorough and the discussions and statistical analyses performed for each of the factors is satisfactory.

1.     As a minor suggestion, a table which would summarize the results in manner of trait, results and the strains reflecting significant difference would help the reader have access to all of the data for an easy comparison.

2.     For the fecundity assay, I am curious if crosses are made between the strains, would that impact the fecundity. For example, females from Neiva (low fecundity across all gonotropic cycles) crosses with various males including the control ROCK would impact this or not. Depending on the geographical distribution, it is not uncommon for Aedes strains of one region to mate with another- impacting vector control and complicating ecological significance further.
